# Dynamic pricing strategies towards strategic consumers under demand learning

Meixian Tang[1], Junfeng Tian[1*], Jinxia Kong[2¤], Zhenzhen Ren[1], Jinsong Tian[3]

1 School of Business Administration, Faculty of Business Administration, Southwestern University of Finance and Economics, Chengdu, China, 2 Smart Hub BU, Cainiao Network Technology Co., Ltd, Hangzhou, China, 3 College of Resource and Environment, Anhui Agricultural University, Hefei, China

¤ 3C Digital Operator, Zhejiang Tmall Technology Co., Ltd, Hangzhou, China
* swufe56@126.com

## Abstract

This study considers a two-stage dynamic pricing framework with demand learning, where a seller sells a finite number of products to strategic consumers in an uncertain market. Based on Bayesian updating, we reveal the applicable conditions of demand learning for dynamic pricing with and without price guarantee. If a price guarantee is not offered in dynamic pricing, demand learning yields marginal benefits only when the initial estimate of prior market size is accurate or overestimated, and prior market size uncertainty is low. Under dynamic pricing with price guarantee, demand learning is effective when the initial estimate is inaccurate, particularly for a prior market size with high uncertainty. However, if the initial estimate is relatively certain, demand learning causes revenue loss by offering price guarantees. Moreover, we also explore the trade-off between dynamic pricing strategies when demand learning is adopted and propose managerial insights. For a product just entering the market and an overestimated market size, dynamic pricing with price guarantee and demand learning dominates. When the product is in the introduction period with an underestimated market size or the product is in the mature period, dynamic pricing without price guarantee and with demand learning is better for weak strategic consumers or under low inventory. Otherwise, dynamic pricing with price guarantee and demand learning is favorable in most cases.

## 1. Introduction

With economic development and consumer demand diversification, most products have uncertain demand, fast replacement, and short life cycles. This creates serious challenges for sellers' revenue management. Generally, sellers such as airlines can adopt dynamic pricing to maximize revenue [1–2]. Indeed, dynamic pricing has been widely used in retail environments in recent years [3–5]. However, because of complex market environments, sellers rarely estimate the true demand curve

**Data availability statement:** The data that support the findings of this paper are available in the manuscript and its appendix.

**Funding:** This work was supported by the National Natural Science Foundation of China (Grant Nos. 71972158, 71571147, and 71901181). The funder had no role in the study design, data collection and analysis, decision to publish, or preparation of the manuscript.

**Competing interests:** The authors have declared that no competing interests exist.

immediately. In this case, dynamic pricing with demand learning needs to be considered; that is, sellers update consumer demand through historical sales data so that the updated demand is close to the actual market demand, and the optimal price can be obtained. Using women's apparel data sold by a fashion cataloger, Fisher [6] shows that forecast demand error by experts is 55%, whereas forecast demand error obtained by learning two-week demand data is only 8%. Thus, dynamic pricing with demand learning is robust and can significantly improve revenue. For instance, Groupon, an American e-commerce firm, learns about demand based on real-time sales data after new products are launched so that it obtains more accurate demand and adjusts prices [7]. Amazon [8], MediaMarkt [9], and Walmart [10] have adopted dynamic pricing with demand learning in daily operations.

However, with numerous shopping experiences, consumers gradually master sellers' pricing rules [11–12]. Consequently, they become "strategic" when purchasing; that is, they not only pay attention to current product price but also estimate future product price and availability, making purchase decisions by the expected utility of immediate purchase and waiting. These consumers are called "strategic consumers" [13–15]. Advanced technology makes access to information more convenient, leading to more strategic consumers. Osadchiy and Bendoly [16] find that 77% of consumers show strategic behavior in intertemporal purchases. This behavior has greatly eroded sellers' revenues. Using numeral studies, Aviv and Pazgal [17] find that sellers will lose approximately 20% of their total revenues if such strategic consumer behavior is ignored.

To relieve the negative impact of strategic consumers and induce them to make early purchases, sellers take various measures. For example, JD.com (NASDAQ: JD), a Chinese e-commerce giant, launches a price guarantee service (also called price protection). Essentially, consumers can apply for price difference compensation if there is a markdown during the price protection period, which is divided into 90, 30, 15, and 7 days according to product attributes. Similarly, Dell [18], The Home Depot [19] and The Good Guys [20] promise a 30-day price guarantee after purchase. Lai et al. [21] believe that offering a price guarantee can reduce consumers' strategic behavior.

However, existing studies primarily emphasize the positive role of price guarantees in mitigating strategic consumers' waiting behavior, while paying limited attention to the potential revenue trade-offs under the coexistence of dynamic pricing and demand uncertainty. When facing strategic consumers, although a price guarantee can incentivize early purchases in the initial stage, sellers often incur considerable compensation costs in the later stages. This creates uncertainty regarding the effectiveness of implementing a price guarantee, further intensifying the seller's hesitation in adopting such a strategy. Moreover, few studies have systematically explored pricing strategy selection under the joint influence of dynamic pricing, demand learning, and strategic consumer behavior. Although some scholars have attempted to incorporate demand learning into dynamic pricing models, most studies neglect the effect of price guarantees, thus failing to reveal whether demand learning remains beneficial in such pricing strategy when faced with strategic consumers, or to clarify

the applicable conditions and constraints under such a setting [22–24]. However, it is particularly important for sellers to find a dynamic pricing strategy that can facilitate demand learning when facing strategic consumers.

Consequently, considering strategic consumers, we formulate two-stage dynamic pricing with and without price guarantee models based on demand learning. We seek to answer the following questions:

(1) Is demand learning always effective for dynamic pricing with and without price guarantee?

(2) What is the impact of demand learning on dynamic pricing with and without price guarantee?

(3) What are the applicable conditions for demand learning for dynamic pricing with and without price guarantee?

(4) How should sellers choose dynamic pricing strategies with demand learning towards strategic consumers in different product sales periods?

### 1.1. Main results and contributions

Our study contributes to the literature in three ways. First, we develop two-stage dynamic pricing models with and without demand learning when a price guarantee is and is not offered to strategic consumers. We set the prior and posterior distributions of market size according to the Bayesian update scheme, and analyze consumers' purchase decisions under constrained inventory. We also prove the existence of conditions for the optimal solutions of pricing models.

Second, we reveal the impact of demand learning on dynamic pricing with and without price guarantee. If there is no offering price guarantee in dynamic pricing, demand learning is slightly beneficial only when the initial estimate is accurate or overestimated, and prior market size uncertainty is low. For dynamic pricing with price guarantee, demand learning can provide significant benefits when the initial estimate is inaccurate, particularly under a highly uncertain prior market size. Conversely, when the initial estimate can be determined relatively accurately, demand learning undermines the effectiveness of a price guarantee. Overall, when considering strategic consumers, demand learning is more suitable for dynamic pricing with price guarantee in most cases.

Finally, we obtain the applicable conditions for implementing dynamic pricing strategies with demand learning based on the product sales period. For the product introduction period and when the market size is overestimated, dynamic pricing with price guarantee and demand learning should be adopted. When (1) the product is in the introduction period and market size is underestimated, or (2) the product is in the mature period, dynamic pricing without price guarantee and with demand learning is suitable only for weak strategic consumers or under low inventory. In all other situations, dynamic pricing with price guarantee and demand learning is preferred.

### 1.2. Paper structure

The remainder of this article proceeds as follows. Section 2 reviews the relevant literature. Section 3 describes the basic problem, decision sequence and methodology, setting of demand learning, and consumer purchase decisions. Section 4 develops several models for the dynamic pricing strategies and explains their computational complexity. Section 5 analyzes the performance of demand learning in dynamic pricing with and without price guarantee. Section 6 discusses the trade-off between dynamic pricing strategies with demand learning. Section 7 extends the models to consider sellers' inventory decisions. Section 8 presents the conclusion, including managerial insights, theoretical implications, limitations, and future research directions. For readability, all proofs are given in the Appendix of the e-companion (S1 Appendix).

## 2. Literature review

Two main research streams are related with this study: dynamic pricing with demand learning; and dynamic pricing with strategic consumers.

## 2.1. Dynamic pricing with demand learning

In recent years, dynamic pricing with demand learning has grown rapidly in operations management. The research on this topic has been reviewed by den Boer [3]. We focus on demand learning using parametric settings [25]. Specifically, sellers estimate the unknown parameters of market size at the beginning of a sale, then learn based on the realized sales data to update it to obtain a new demand function or distribution, and finally, set the price for the next period. Regarding the demand distribution, Aviv and Pazgal [26] first adopt the Poisson-Gamma distribution to modify the demand model through a *Bayesian update rule*. Subsequent studies on demand learning primarily follow this approach. For instance, Aviv and Pazgal [27] and Lin [28] develop a dynamic pricing model with a finite horizon and unknown consumer arrival rate, and suggest a heuristic pricing policy in which sellers continuously learn the consumer arrival rate using real-time sales data. Sen and Zhang [29] examine multistage optimal pricing using the Poisson-Gamma distribution to learn the consumer arrival rate and summarize the conditions of dynamic pricing with demand learning through two-stage numerical studies. Araman and Caldentey [30] assume that market size is unknown but has a prior distribution, and propose that it can be updated using sales time and data to obtain the posterior distribution. Other learning methods have also become popular in recent years. Broder and Rusmevichientong [31] examine dynamic pricing under a general parameter choice model and propose a price policy based on observed consumer purchasing decisions using maximum likelihood estimation. Besbes and Zeevi [32] show that demand learning reduces the deviation between a seller's basic price and the real price when these are significantly different. Cao et al. [33], den Boer and Keskin [34] and Wang et al. [22] consider a dynamic pricing problem with demand learning and reference effects.

Notably, while all aforementioned studies focus on incorporating demand learning into dynamic pricing to set a multistage price for a seller, they neither consider consumers' strategic behavior nor dynamic pricing with price guarantee. Meanwhile, this study examines the impact of demand learning on dynamic pricing with and without price guarantee when facing strategic consumers.

## 2.2. Dynamic pricing towards strategic consumers

Since Coase [35] proposes the "Coase Conjecture" that a monopolistic seller only sells products at marginal cost when facing rational consumers waiting for a markdown, research on strategic consumers is rapidly growing. Wei and Zhang [36] provide a detailed literature review on consumers' strategic behavior. Besanko and Winston [37] set up a game theory model to analyze how strategic consumers affect a monopolistic seller's price and profit, highlighting that the price for strategic consumers is always lower than that for myopic consumers. Nair [38] empirically analyzes the substantial impact of strategic consumers on sellers' prices and revenues using US gaming market data, and provides an optimal pricing policy.

Sellers often use dynamic pricing when faced with strategic consumers. Aviv and Pazgal [17] believe that strategic consumer purchase decisions follow a threshold policy, meaning that goods are bought right away if the valuation is higher than the threshold. They show that dynamic pricing is inferior to price commitments for sellers. Cachon and Swinney [39] indicate that, consumers' strategic behavior is effectively controlled through a quick response strategy that embeds dynamic pricing and inventory replenishment. Chen et al. [40] design two dynamic pricing strategies (markdown and markup) under centralized and decentralized systems when considering the reference price effect and strategic consumers. Li and Chen [41] further explore how e-commerce platforms and retailers trade off with two-stage discount pricing strategies when facing strategic consumers. However, although classic dynamic pricing offers price flexibility, it does not significantly alleviate consumers' strategic behavior. Therefore, many sellers offer price compensation based on dynamic pricing, which means that sellers promise to compensate early high-priced consumers for the price difference when the product price decreases. Illustrative examples include price guarantee or price matching.

This strategy can help maintain dynamic pricing flexibility and induce strategic consumers to make early purchases. Png [42] explores the performance of most-favored-customer (MFC) protection (price guarantee) under strategic consumers when the initial inventory is exogenous, finding that MFC protection is favored when initial inventory is large. Lai

et al. [21] examine inventory and price decisions about implementing posterior price matching when facing strategic and myopic consumers. The authors find that posterior price matching eliminates strategic consumers' waiting behavior and is dominant under certain conditions. Yan and Ke [43] propose two pricing strategies for strategic consumers—posterior and delayed posterior price matching—and obtain the applicable conditions for each strategy. Shum et al. [44] compare dynamic pricing, price commitment, and price matching under strategic consumers, exploring the impact of technology advancement and production learning on these strategies. Zhao et al. [45] explore the trade-off between dynamic pricing and price matching under reference price effects and strategic consumers, and obtain the applicable conditions for price matching. Recently, dynamic pricing with price guarantee has been used for advanced selling [46–48] and omni-channel purchasing [49].

It should be noted that price matching is also used in competitive environments [50–55]. Here, price matching refers to the seller's promise to match a lower price if competitors sell the same product. Hence, in our study, *price guarantee* is specifically represented as the seller's price commitment to compensate consumers who purchase at a high price in the second stage of dynamic pricing.

Extant research on strategic consumers often assume that demand is deterministic or that the demand distribution is known. Few studies on dynamic pricing consider demand learning towards strategic consumer behavior. Levina et al. [56] propose an aggregation algorithm for dynamic pricing with demand learning; numerical simulations suggest models with demand learning that perform more robustly and better than models without learning under strategic consumers. However, Aviv et al. [57] demonstrate that sellers' demand learning can be worse off when facing strategic consumers.

### 2.3. Differences and contributions to the literature

There are three differences between our paper and existing literature. First, as far as we know, currently there is no study considering the interaction among demand learning, dynamic pricing without and with price guarantee, and strategic purchase. Therefore, to fill this research gap, we examine the impact of demand learning on dynamic pricing with and without price guarantee for strategic consumers. Second, we consider two-sided learning [58–59] in which both the seller and consumers update their prior market size. Papanastasiou and Savva [58] emphasize that both the seller and consumers share a common prior belief over ex-ante unknown quality, and update the prior belief according to Bayesian rule after observing previous reviews. Hu et al. [60] argue that the seller and consumers simultaneously learn the market size in the second stage when the seller discloses the sales in the first stage. Finally, we set the perfect information model as the upper bound to measure the effectiveness of demand learning for the two dynamic pricing alternatives. Table 1 shows the comparison between our study and the relevant literature.

In summary, we contribute to the literature on dynamic pricing with demand learning by demonstrating the applicable conditions of demand learning under dynamic pricing with and without price guarantee, and how to choose dynamic pricing strategies with demand learning.

## 3. Model setup

This section introduces the problem statement, decision sequence and methodology, demand learning approach, and consumer purchasing decisions.

### 3.1. Basic statement

We assume that a monopolistic seller sells $Q$ units of a product to strategic consumers within a finite horizon. The seller has an opportunity to alter the price at a given time and the sales horizon is divided into two stages. Two alternatives can be chosen for the seller's dynamic pricing: (a) *dynamic pricing without price guarantee*, which means that only the first-stage price $p_1$ is announced and $p_2$ will be set at the beginning of the second stage; and (b) to counter consumers' strategic behavior, *dynamic pricing with price guarantee* is favored by many sellers, such as Tmall.com, JD.com, Dell, The

**Table 1. A comparison between our work and relevant research.**

| Authors | Demand Learning | Strategic consumers | Dynamic pricing without price guarantee | Dynamic pricing with price guarantee | Decision variables |
|---|---|---|---|---|---|
| Png [42] | × | √ | √ | √ | Price |
| Lin [28] | √ | × | √ | × | Price |
| Aviv and Pazgal [17] | × | √ | √ | × | Price |
| Sen and Zhang [29] | √ | × | √ | × | Price |
| Lai et al. [21] | × | √ | √ | √ | Price and inventory |
| Talebian et al. [61] | √ | × | √ | × | Price and assortment |
| Papanastasiou and Savva [58] | × | √ | √ | × | Price |
| Shum et al. [44] | × | √ | √ | √ | Price |
| Aviv et al. [57] | √ | √ | √ | × | Price |
| Liu et al. [62] | √ | × | √ | × | Price and inventory |
| den Boer and Keskin [34] | √ | × | √ | × | Price |
| Wang et al. [22] | √ | × | √ | × | Price |
| Yang et al. [24] | √ | × | √ | × | Price |
| **Our work** | √ | √ | √ | √ | Price and inventory |

Note: "√" represents research content, and "×" represents non-research content.

Home Depot, and The Good Guys. If $p_2 < p_1$, the seller will provide a full price difference compensation once consumers submit an application for it. Therefore, under dynamic pricing with price guarantee, consumers may receive additional compensation, but the seller incurs the corresponding costs.

To ensure tractability, the marginal cost of the product is set to zero [60,63] and replenishment is not allowed during sales period. If products remain at the end of the horizon, the salvage value is zero [63].

Assume that the number of strategic consumers interested in the product is random, and that consumers enter the market in a short time at the beginning of the sales process. Similar to mainstream studies [29,57,61], the number of consumers in the sales period is drawn from a Poisson distribution with a mean value of $\Lambda$; $\Lambda$ also represents the market size. This assumption is reasonable. In practice, for example, the number of arrivals at the market and stores [64–65], or visiting online stores and websites [66–67] is represented by a Poisson distribution. However, the mean value of Poisson distribution $\Lambda$ is unknown and only has a prior Gamma distribution formed by the seller. Here, the Gamma distribution is chosen because it is a conjugate prior distribution of the Poisson distribution [68], which is consistent with the classical studies [26–29,57,61]. After observing actual sales in the first stage, a posterior distribution of $\Lambda$ is obtained. This is called demand learning and is stated specifically in Section 3.3.

Each consumer buys at most one product and consumers' valuation $v$ is heterogeneous, which is independently drawn from a uniform distribution on $[0, 1]$ [44,57]. The net utility is non-negative when consumers purchase goods at each stage; otherwise, they exit the market and obtain a net utility of zero. Moreover, all strategic consumers make purchasing decisions by comparing the expected utility of the two stages. If they choose to delay purchasing, the product's future net utility will be discounted over time by a factor $\delta \in (0, 1]$. $\delta$ measures how patient or strategic consumers are. The higher the value, the more strategic or patient consumers. $\delta$ also represents the durability of the goods. A higher $\delta$ implies a more durable goods with greater future value [39,58]. Consequently, consumers with the valuation $v$ have a net utility $u_2 = \delta(v - p_2)$ in the second stage [44,45,58]. Note that when consumers make purchasing decisions in the first stage, they have no idea of the second-stage price $p_2$. Therefore, consumers form a belief about the second-stage price, which is represented by $\hat{p}_2(q)$. Let $q = (Q - x_1)^+$ denote the inventory in the second stage and $x_1$ be the first-stage sales data. Assume that the belief $\hat{p}_2(q)$ is obtained according to the Rational Expectation Equilibrium (REE) first introduced by Muth

[69]. REE believes that if decision-makers are completely rational and information is fully utilized, their expectations of future outcomes are consistent with the actual results. Later it is widely used in many studies on strategic consumers, such as [44,70–72]. Here, rational expectations emphasize that consumers form correct beliefs about the seller's pricing and other consumer behaviors to make purchasing decisions; and that the seller optimizes pricing subject to the correct belief about consumers.

Obviously, owing to the dissimilar expected utility, we thus believe that strategic consumers' purchasing decisions are different under dynamic pricing with and without price guarantee. Because the seller considers whether to update the prior market size by demand learning after observing actual first-stage sales to consumers, the effectiveness and performance difference in the two classes of dynamic pricing needs to be confirmed. Consequently, we examine four dynamic pricing strategies, represented by $K = \{DN, DL, GN, GL\}$. These represent dynamic pricing without price guarantee and demand learning (**Strategy DN**); dynamic pricing without price guarantee and with demand learning (**Strategy DL**); dynamic pricing with price guarantee and without demand learning (**Strategy GN**); and dynamic pricing with price guarantee and demand learning (**Strategy GL**).

The basic notations are summarized in Table 2.

### 3.2 Decision sequence and methodology

The decision sequence of our model is shown in Fig. 1.

The methodology is as follows: (1) Based on utility function theory, the seller can obtain a two-stage demand segment and dynamic pricing models are formulated. (2) Using backward induction, we determine the optimal price for the seller and first-stage purchase threshold for consumers. (3) For the two dynamic pricing alternatives, the solution of their respective perfect information models is used as a benchmark to evaluate the performance of demand learning and explore the applicable conditions for adopting demand learning. (4) By comparing the expected revenue, the trade-off between dynamic pricing strategies with demand learning is analyzed.

**Table 2. Notations.**

| Notation | Description/definition |
|---|---|
| **Parameters** | |
| $\Lambda$ | Average market size (the market size for short) |
| $\alpha, \beta$ | Parameters of the Gamma distribution |
| $v$ | Consumer's valuation for production |
| $\delta$ | Discount factor of consumer's utility ($0 < \delta \leq 1$) |
| $Q$ | Initial inventory in base model |
| $x_1$ | Realized sales data in the first stage |
| $K = \{DN, DL, GN, GL\}$ | Dynamic pricing strategy |
| $D_i^K, i = \{1, 2\}$ | Estimated demand distribution in the $i$th stage in strategy $K$ |
| $U_i^K, i = \{1, 2\}$ | Consumer's expected surplus in the $i$th stage in strategy $K$ |
| $u_i^K, i = \{1, 2\}$ | Consumer's net utility in the $i$th stage in pricing $K$ |
| $f_i^K(\cdot), i = \{1, 2\}$ | Consumer's perceived fill rate in the $i$th stage in strategy $K$ |
| $R_2^K$ | Expected revenue in the second stage in strategy $K$ |
| $R^K$ | Total expected revenue of strategy $K$ |
| **Decision variables** | |
| $p_i^K, i = \{1, 2\}$ | Price in the $i$th stage in strategy $K$ |
| $Q$ | Initial inventory in extended model |

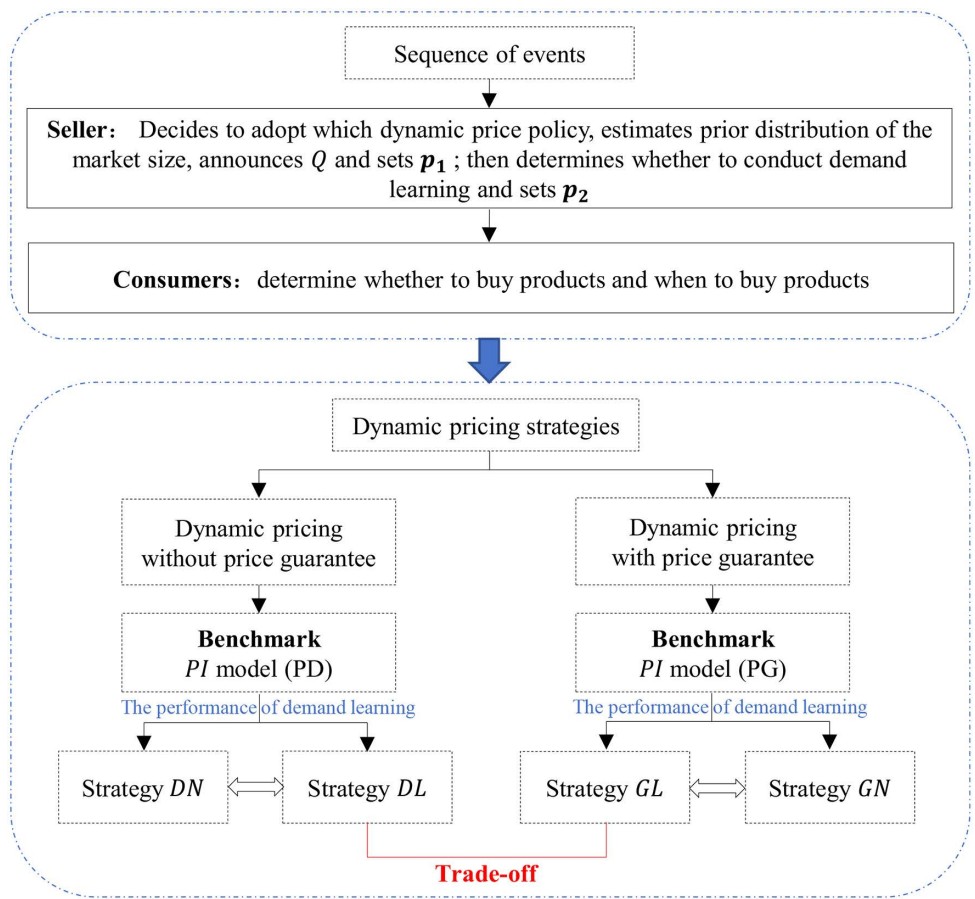

**Fig 1. Sequence of events.** The seller first decides to adopt dynamic pricing with or without price guarantee, then estimates the prior distribution of the market size, announces the initial inventory level Q, and finally posts the price p₁. Consumers observe the released information (offering of price guarantee, Q, and p₁), form the belief p̂₂(q) about the second-stage price, and make purchase decisions. The product is purchased in the first stage if and only if (a) the first-stage expected surplus is non-negative; and (b) the first-stage expected surplus is no less than that in the second stage. Subsequently, after observing the first-stage sales x₁, the seller decides whether to conduct demand learning to update the prior distribution of the market size and determines p₂. The remaining consumers who do not purchase in the first stage make purchase decisions based on the given p₂. In summary, the seller decides which pricing strategy and two-stage price to adopt, while consumers determine whether and when to buy the products.

### 3.3. Demand learning

All aforementioned parameters $\{Q, x_1, \alpha, \beta, \delta\}$ are assumed to be common knowledge for the seller and consumers. This assumption is often used in studies on strategic consumers. For example, Papanastasiou and Savva [58] mention that the seller and consumers hold the same prior distribution over an ex-ante unknown quality.

The Poisson-Gamma model has been widely adopted for demand learning and forecasting [26–29,61,73–75]. Aviv et al. [57] state that the Poisson-Gamma model is reasonable as it allows the specification of large demand uncertainty scenarios. The conditional probability distribution of demand at each stage follows a Poisson distribution with the mean $\Lambda\Psi_i(p_i)$. Here, $\Psi_i(p_i)$ is a deterministic function of the price $p_i$ at each stage and usually expressed as the purchase probability at the price $p_i$. $\Lambda$ is a random variable which indicates the market size and has a prior distribution $\Gamma(\alpha, \beta)$, where the parameters $\alpha$ and $\beta$ are formed by the seller through market research or empirical knowledge. Here, $E(\Lambda) = \frac{\alpha}{\beta}$, $Var(\Lambda) = \frac{\alpha}{\beta^2}$ and $CV(\Lambda) = \frac{\sqrt{Var(\Lambda)}}{E(\Lambda)} = \frac{1}{\sqrt{\alpha}}$ for the prior distribution. Since the actual sales data can reflect part of the market

information, the seller has an opportunity to learn sales to update the prior distribution of $\Lambda$. When the seller sets the price $p_1$ and observes sales volume $x_1$ in the first stage, the posterior distribution of $\Lambda$ can be obtained according to the Bayesian update scheme. In the Poisson-Gamma model, we obtain the following Lemma 1 for the unconditional distribution of the estimated demand at the beginning of the two stages and posterior distribution of $\Lambda$ updated by demand learning (see Appendix A for the proof).

*Lemma 1. When the deterministic function $\Psi_i(p_i)$, $(i = 1, 2)$ is given, the demand distributions of the two stages and posterior distribution of $\Lambda$ are as follows.*

(i) *The estimated demand $D_1$ at the beginning of the first stage follows a negative binomial distribution with the parameter $\left(\alpha, \frac{\beta}{\beta + \Psi_1(p_1)}\right)$, whose mean is $\frac{\alpha \Psi_1(p_1)}{\beta}$.*

(ii) *After observing the real demand $x_1$ in the first stage, the posterior distribution of $\Lambda$ is updated to a Gamma distribution with the parameter $(\alpha + x_1, \beta + \Psi_1(p_1))$.*

(iii) *After observing the real demand $x_1$ in the first stage, the estimated demand $D_2$ at the beginning of the second stage follows a negative binomial distribution with the parameter $\left(\alpha + x_1, \frac{\beta + \Psi_1(p_1)}{\beta + \Psi_1(p_1) + \Psi_2(p_2)}\right)$, and its mean is $\frac{(\alpha + x_1)\Psi_2(p_2)}{\beta + \Psi_1(p_1)}$.*

Lemma 1(ii) shows that the posterior distribution of the market size depends on the prior information (demand estimation) and observed sales data $x_1$ (demand learning). Besides, according to statistical knowledge, the coefficient of variation on market size after demand learning is smaller than that of the initial prior distribution $(CV(\Lambda|x_1) = \frac{1}{\sqrt{\alpha + x_1}} < CV(\Lambda) = \frac{1}{\sqrt{\alpha}})$. This further shows that demand learning can reduce the uncertainty of $\Lambda$. Lemma 1(i) and 1(iii) show that the estimated demand distribution of the two stages is a negative binomial distribution, which is obtained by calculating the Poisson-Gamma composite distribution.

However, without demand learning, the unconditional distribution of the estimated demand for each stage is also a negative binomial distribution, expressed as $D_i \sim NB(\alpha, \frac{\beta}{\beta + \Psi_i(p_i)})$. Intuitively, the estimated unconditional second-stage demand $D_2$ is independent of the first-stage sales volume.

### 3.4. Consumer purchase decisions

When consumers arrive at the market, they need to decide whether and when to buy products according to their valuation $v$ and announced information. However, consumers not only consider the seller's future second-stage price but also compete against *other* consumers' purchase behavior because this can affect product availability in the two stages. Specifically, a consumer with valuation $v$ makes a purchase decision by comparing the expected surplus of immediate purchase and that of waiting for the purchase.

Due to demand uncertainty and limited inventory, consumers must consider the possibility of available items at each stage. To describe this, we define the likelihood that a unit is allocated to consumers at each stage as *the perceived fill rate* [76], denoted as $f_i^K(\cdot)$, $i = \{1, 2\}$ and $K = \{DN, DL, GN, GL\}$ (see Table 2). Thus, the expected surplus at each stage is equal to the net utility multiplied by the perceived fill rate at the current stage. Specifically, given $K$, $Q$, $p_1^K$ and a consumer's belief of the second-stage price $\hat{p}_2^K(q)$, the first-stage expected surplus represents $U_1^k = f_1^K(Q) \cdot u_1^K$, while the second-stage expected surplus is $U_2^K = \delta \sum_{x_1}^Q f_2^K(q) \cdot u_2^K \cdot \Pr(D_1^K = x_1)$. And the expressions for the two-stage expected surplus are summarize in Table 3 (see Appendix A for details).

Considering other consumers' purchase behavior and the anticipated second-stage price, we obtain Theorem 1 (see Appendix A for the proof).

**Theorem 1.** *For a given $K$, $Q$, and $p_1^K$ and the belief of the second-stage price $\hat{p}_2^K(q)$, a consumer with valuation $v$ makes purchase decisions using the following threshold policy.*

(i) *For $K = \{DN, DL\}$, $\bar{v}_1^K = v_1^K(p_1^K)$, in which $v_1^K(p_1^K) \in [p_1^K, 1]$, is the unique solution when $U_1^K = U_2^K$. If $v \geq \bar{v}_1^K$, the consumer chooses to buy a product in the first stage; otherwise, the consumer waits for the second stage.*

**Table 3. The two-stage expected surplus for different dynamic pricing strategies.**

| Dynamic pricing strategies | Expected surplus |
|---|---|
| No offering price guarantee ($K = \{DN, DL\}$) | $U_1^K = f_1^K(Q) \cdot (v - p_1^K)$ |
| | $U_2^K = \delta \sum_{x_1=0}^{Q} f_2^K(q) \cdot (v - \hat{p}_2^K(q)) \cdot \Pr(D_1^K = x_1)$ |
| Offering price guarantee ($K = \{GN, GL\}$) | $U_1^K = f_1^K(Q) \cdot \left[ v - p_1^K + \delta \sum_{x_1=0}^{Q} \left( p_1^K - \hat{p}_2^K(q) \right)^+ \cdot \Pr\left(D_1^K = x_1\right) \right]$ |
| | $U_2^K = \delta \sum_{x_1=0}^{Q} f_2^K(q) \cdot \left(v - \hat{p}_2^K(q)\right) \cdot \Pr\left(D_1^K = x_1\right)$ |

(ii) *For $K = \{GN, GL\}$, $\bar{v}_1^K = p_1^K$. If $v \geq \bar{v}_1^K$, the consumer chooses to buy a product in the first stage; otherwise, the consumer waits for the second stage.*

(iii) *Finally, $\bar{v}_2^K = \hat{p}_2^K(q)$. The consumer buys a product in the second stage if and only if $v \in [\bar{v}_2^K, \bar{v}_1^K)$.*

Theorem 1 implies that a consumer with high valuation chooses to buy products in the first stage, while a consumer with low valuation delays purchasing. Theorem 1(i) shows that for dynamic pricing without price guarantee, regardless of demand learning, the first-stage purchase threshold for a consumer is higher than the first-stage price. However, as shown in Theorem 1(ii), for dynamic pricing with price guarantee, a consumer purchases in the first stage as long as $v$ is higher than the first-stage price, instead of considering the second-stage expected surplus. That is, offering a price guarantee can eliminate strategic consumers' waiting behavior by compensating for price differences.

According to Theorem 1, we obtain Proposition 1 to describe the equilibrium in consumer competition (see Appendix A for the proof).

*Proposition 1. Given $K$, $Q$, and $p_1^K$ and the belief of the second-stage price $\hat{p}_2^K(q)$, there is a unique Nash equilibrium $\bar{v}_1^K$ in the competition of consumers in the first stage; that is, all consumers conduct the same purchase threshold in the first stage.*

For the perceived fill rate at a given stage, we consider the following model setup: First, consumers estimate the perceived fill rate based on currently available inventories and the number of *other* consumers. Assume that from any individual consumer's perspective, the demand distribution of *other* consumers is consistent with that of all consumers [57,76]. If the current demand is less than the inventory, consumers obtain a unit item with probability one; otherwise, all available units are randomly allocated among consumers who buy the product with an equal probability. Furthermore, the seller publishes sales data $x_1$ as public information. An illustrative example is the sales data of products posted by sellers on main retail platforms in China, such as Tmall.com and Taobao.com. Therefore, we believe that consumers can observe public information $x_1$ at the beginning of the second stage and update the prior distribution of the market size. This is reasonable because the updated market size affects the perceived fill rate in the second stage. Suppose that the demand learning of the seller and consumers is homogenous [58,60]. Consequently, the perceived fill rates $f_1^K(Q)$ and $f_2^K(q)$ (if $q \geq 1$) in two stages are expressed as follows.

$$f_1^K(Q) = \sum_{x_1=0}^{Q-1} P\left(D_1^K = x_1\right) + \sum_{x_1=Q}^{\infty} \frac{Q}{x_1+1} P\left(D_1^K = x_1\right)$$
$$= \left(1 - \frac{Q}{E\left(D_1^K\right)}\right) \sum_{x_1=0}^{Q-1} P\left(D_1^K = x_1\right) + \frac{Q}{E\left(D_1^K\right)} \left[1 - P\left(D_1^K = Q\right)\right] \tag{1}$$

$$f_2^K(q) = \sum_{x_2=0}^{q-1} P\left(D_2^K = x_2\right) + \sum_{x_2=q}^{\infty} P\left(D_2^K = x_2\right) \cdot \frac{q}{x_2+1}$$
$$= \left(1 - \frac{q}{E\left(D_2^K\right)}\right) \sum_{x_2=0}^{q-1} P\left(D_2^K = x_2\right) + \frac{q}{E\left(D_2^K\right)} \left[1 - P\left(D_2^K = q\right)\right] \tag{2}$$

In Eqs. (1) and (2), according to Lemma 1 and Theorem 1, when demand learning is considered, the average values of estimated demand $D_1^K$ and $D_2^K$ in the two stages are represented as $E(D_1^K) = (1 - \bar{v}_1^K)\frac{\alpha}{\beta}$ and $E(D_2^K) = (\bar{v}_1^K - p_2^K)\frac{\alpha + x_1}{\beta + 1 - \bar{v}_1^K}$, respectively.

## 4. Seller's dynamic pricing strategies for demand learning

This section develops several two-stage models with and without demand learning when the seller employs dynamic pricing with and without price guarantee. Accordingly, as stated in Section 3.1, four dynamic pricing strategies are investigated.

In dynamic pricing without price guarantee, the seller first announces the price $p_1$, and subsequently, consumers make the first-stage purchase decisions. At the beginning of the second stage, the seller decides $p_2$ according to previous sales, and the consumers remaining in the market make the second-stage purchase decisions.

Dynamic pricing with price guarantee operates as follows: the seller determines $p_1$ before the sale begins, and then sets $p_2$ based on the remaining inventory at the end of the first stage. Additionally, if there is a markdown ($p_2 < p_1$), the seller compensates consumers who purchase early at a high price ($p_1 - p_2$). Suppose that all consumers who purchase at a high price apply for compensation during the price protection period.

### 4.1 Strategy DN: Without price guarantee and demand learning

Dynamic pricing without price guarantee and demand learning (Strategy $DN$) means that the seller believes that the prior market size distribution is correct. The first-stage sales data $x_1$ are only used to adjust the second-stage price, rather than for updating the prior distribution of market size $\Lambda$. Consequently, the distribution of the market size remains the same $\Lambda \sim \Gamma(\alpha, \beta)$ for the entire sales process. At the time, consumers with a valuation $v$ higher than $v_1^{DN}(p_1^{DN})$ choose to buy in the first stage, while other consumers choose to wait for the second stage. Therefore, the demand distribution of the two stages can be written as $D_1^{DN} \sim NB\left(\alpha, \frac{\beta}{\beta + 1 - v_1^{DN}(p_1^{DN})}\right)$ and $D_2^{DN} \sim NB\left(\alpha, \frac{\beta}{\beta + v_1^{DN}(p_1^{DN}) - p_2^{DN}}\right)$, respectively.

The expected revenue in the second stage is $R_2^{DN} = p_2^{DN} \cdot E_{D_2^{DN}}[\min(D_2^{DN}, q)]$; hence, the seller obtains the optimal price $p_2^{DN*} = \text{argmax}_{p_2^{DN}} R_2^{DN}$. At the beginning of the first stage, the seller sets the optimal price $p_1^{DN*} = \text{argmax}_{p_1^{DN}}\{R^{DN} = p_1^{DN} \cdot E_{D_1^{DN}}\left[\min\left(D_1^{DN}, Q\right)\right] + E_{D_1^{DN}}(R_2^{DN*})\}$.

### 4.2 Strategy DL: Without price guarantee and with demand learning

Dynamic pricing without price guarantee and with demand learning (Strategy $DL$) represents that at the beginning of the second stage, the seller and consumers update the prior market size based on the first-stage realized sales data $x_1$, and subsequently, the seller sets the second-stage price. Based on Proposition 1, consumers whose valuation $v$ is greater than $v_1^{DL}(p_1^{DL})$ buy products in the first stage. When the remaining consumers in the market observe the second-stage price $p_2^{DL}$, they purchase the products if and only if $v - p_2^{DL} \geq 0$. According to Lemma 1 and Proposition 1, the estimated demand distribution of the second stage is $D_2^{DL} \sim NB\left(\alpha + x_1, \frac{\beta + 1 - v_1^{DL}(p_1^{DL})}{\beta + 1 - p_2^{DL}}\right)$. Thus, the seller's second-stage expected revenue is expressed as follows:

$$R_2^{DL} = p_2^{DL} \cdot E_{D_2^{DL}}\left[\min\left(D_2^{DL}, q\right)\right] \tag{3}$$

where $E_{D_2^{DL}}[\min(D_2^{DL}, q)] = q - \sum_{x=0}^{q-1}(q - x)P(D_2^{DL} = x)$, indicating the expected number of items sold in the second stage.

Considering the first-stage purchase behavior of consumers and seller's second-stage pricing decision, we further explore the seller's first-stage pricing problem. The seller's total expected revenue is expressed as follows:

$$R^{DL} = E_{D_1^{DL}}\left[p_1^{DL} \cdot \min\left(D_1^{DL}, Q\right) + R_2^{DL*}\right] = p_1^{DL} \cdot E_{D_1^{DL}}\left[\min\left(D_1^{DL}, Q\right)\right] + E_{D_1^{DL}}\left(R_2^{DL*}\right) \tag{4}$$

where $D_1^{DL} \sim NB\left(\alpha, \frac{\beta}{\beta+1-v_1^{DL}(p_1^{DL})}\right)$. The first term of Eq. (4) is the expected revenue in the first stage and second term is the optimal expected revenue in the second stage. We explain the optimal solutions using Theorem 2 (see Appendix A for the proof).

**Theorem 2.** *(i) For the optimal first-stage price $p_1^{DL*} = argmaxR^{DL}$, the optimal first-stage purchase threshold $v_1^{DL*}\left(p_1^{DL*}\right)$ satisfies the implicit equation $f_1^{DL}(Q) \cdot \left(v_1^{DL*}\left(p_1^{DL*}\right) - p_1^{DL*}\right) = \delta \sum_{x_1=0}^{R_1^{DL} Q} \left(f_2^{DL}(q) \cdot \left(v_1^{DL*}\left(p_1^{DL*}\right) - p_2^{DL*}\right)\right) \cdot Pr(D_1^{DL} = x_1)$, and $v_1^{DL*}\left(p_1^{DL*}\right)$ is increasing in $\delta$.*

*(ii) The optimal second-stage price $p_2^{DL*}$ is unique for any realized $x_1$, and $p_2^{DL*} \in [0, v_1^{DL*}\left(p_1^{DL*}\right)]$.*

Theorem 2(i) demonstrates that consumers' first-stage purchase thresholds increase with the strategic behavior level. That is, consumers are more strategic and are willing to wait. Theorem 2(ii) indicates the existence and uniqueness of $p_2^{DL*}$. After obtaining the consumer purchasing condition $v_1^{DL*}\left(p_1^{DL*}\right)$ and actual sales $x_1$ in the first stage, the seller can determine the optimal price $p_2^{DL*}$ by maximizing Eq. (3).

## 4.3 Strategy GN: With price guarantee and without demand learning

In dynamic pricing with price guarantee and without demand learning (Strategy *GN*), the seller offers compensation to consumers who have purchased early at a high price in the second stage; however, the seller believes that the prior market size is accurate and does not consider demand learning. Therefore, the distribution of the market size in the two stages is consistent, and the estimated demands are $D_1^{GN} \sim NB(\alpha, \frac{\beta}{\beta+1-p_1^{GN}})$ and $D_2^{GN} \sim NB(\alpha, \frac{\beta}{\beta+p_1^{GN}-p_2^{GN}})$, respectively. When the price $p_1^{GN}$ and realization $x_1$ of sales in the first stage are known, the expected revenue in the second is described as $R_2^{GN} = p_2^{GN} \cdot E_{D_2^{GN}}\left[\min\left(D_2^{GN}, q\right)\right] + x_1(p_1^{GN} - p_2^{GN})^+$. The second term is the compensation that the seller pays to the consumers who purchased in the first stage. We set the optimal price for the second stage $p_2^{GN*} = argmax_{p_2^{GN}} R_2^{GN}$. Then, the seller can set the optimal price of the first stage by $p_1^{GN*} = argmax_{p_1^{GN}} R^{GN} = p_1^{GN} \cdot E_{D_1^{GN}}\left[\min\left(D_1^{GN}, Q\right)\right] + E_{D_1^{GN}}(R_2^{GN*})$.

## 4.4 Strategy GL: With Price Guarantee and Demand Learning

Similar to Section 4.2, dynamic pricing with price guarantee and demand learning (Strategy *GL*) shows that in the second stage, the seller and consumers make decisions by updating their prior distribution of the market size $\Lambda$ based on the first-stage sales data. In addition, as the price guarantee is offered, consumers with a valuation $v$ higher than $p_1^{GL}$ choose to purchase in the first stage. According to Lemma 1 and Proposition 1, the updated estimated demand distribution of the second-stage $D_2^{GL} \sim NB\left(\alpha + x_1, \frac{\beta+1-p_1^{GL}}{\beta+1-p_2^{GL}}\right)$. Thus, the expected revenue function in the second stage is as follows:

$$R_2^{GL} = p_2^{GL} \cdot E_{D_2^{GL}}\left[\min\left(D_2^{GL}, q\right)\right] - x_1\left(p_1^{GL} - p_2^{GL}\right)^+ \tag{5}$$

The first term of Eq. (5) represents the expected revenue in the second stage, while the second term shows the compensation that the seller pays to the consumers who purchased in the first stage.

The seller's total expected revenue is expressed as follows:

$$R^{GL} = E_{D_1^{GL}}\left[p_1^{GL} \cdot \min\left(D_1^{GL}, Q\right) + R_2^{GL*}\right] = p_1^{GL} \cdot E_{D_1^{GL}}\left[\min\left(D_1^{GL}, Q\right)\right] + E_{D_1^{GL}}(R_2^{GL*}) \tag{6}$$

In Eq. (6), $D_1^{GL} \sim NB(\alpha, \frac{\beta}{\beta+1-p_1^{GL}})$, the first term is the expected revenue of the first stage, and the second term is the optimal expected revenue of the second stage. We describe the optimal solutions using Theorem 3 (see Appendix A for the proof).

**Theorem 3.** *(i) After obtaining the optimal first-stage price $p_1^{GL*} = argmax_{p_1^{GL}} R^{GL}$ according to Eq. (6), the optimal second-stage price $p_2^{GL*}$ is unique for any realized $x_1$ and $p_2^{GL*} \in [0, p_1^{GL*}]$.*

*(ii) The optimal total expected revenue $R^{GL*}$ is independent of $\delta$.*

For Strategy *GL*, Theorem 3(i) indicates the existence and uniqueness of the optimal price in the second stage as well as its bounds. Obviously, $p_2^{GL*}$ is a function of the optimal first-stage price $p_1^{GL*}$ and realized sales data $x_1$. When both are known, the seller can determine the optimal price $p_2^{GL*}$ by maximizing Eq. (5). Theorem 3(ii) shows that the level of strategic consumer behavior does not affect a seller's optimal expected revenue.

### 4.5 Computational complexity on pricing models

Note that although we provide the optimality conditions of the above models, owing to their complexity, it is impossible to obtain analytical expressions for the seller's two-stage price and expected revenue, consistent with mainstream studies [17,39,57,72,76]. The specific reasons for these two-stage pricing models are as follows.

(1) The two-stage estimated demand is a negative binomial distribution with respect to prices, making it impossible to calculate the derivative.

(2) The two-stage expected revenue function is a non-polynomial function of prices. Consequently, an explicit expression for prices cannot be obtained by solving the equation under first-order optimal conditions.

(3) For dynamic pricing models without price guarantee ($K = \{DN, DL\}$), an explicit expression for $\bar{v}_1^K$ cannot be obtained because the perceived fill rates $f_1^K(Q)$ and $f_2^K(q)$ are nonlinear implicit functions of $\bar{v}_1^K$, which contains a negative binomial distribution.

Based on the above factors, the closed-form or analytic solution for the optimal two-stage prices cannot be obtained because the first-order optimal condition for the seller's expected revenue is a transcendental equation containing a non-continuous function.

Therefore, following the literature [17,39,57,72,76], we present several numerical studies to explore the several unsolved issues. For example, how effective is demand learning in dynamic pricing with/without price guarantee and how do some important parameters (e.g., the initial estimate and uncertainty of prior market size) affect the performance of demand learning? What are the application conditions for demand learning under dynamic pricing with/ without price guarantee? How can strategic consumers select dynamic pricing strategies based on demand learning?

### 5. Performance of demand learning in two dynamic pricing alternatives

To compare the performance of models with and without demand learning, we set a *perfect (full or complete) information model* (hereinafter, *PI* model) [26,28,29,61,77]. That is, the seller can know the true market size as $\Lambda = \lambda$. In the *PI* model of dynamic pricing without price guarantee (denoted as PD model), the demand distributions in the two stages follow a Poisson distribution, denoted as $D_1^{PD} \sim P[(1 - v_1^{PD}(p_1^{PD}))\lambda]$ and $D_2^{PD} \sim P[(v_1^{PD}(p_1^{PD}) - p_2^{PD})\lambda]$, respectively. The expected revenue functions are as follows:

$$\begin{cases} R_2^{PD} = p_2^{PD} \cdot E_{D_2^{PD}}\left[\min\left(D_2^{PD}, q\right)\right] \\ R^{PD} = p_1^{PD} \cdot E_{D_1^{PD}}[\min(D_1^{PD}, Q)] + E_{D_1^{PD}}(R_2^{PD*}) \end{cases} \tag{7}$$

Similarly, for the *PI* model of dynamic pricing with price guarantee (denoted as PG model), the demand distributions in the two stages are $D_1^{PG} \sim P[(1 - p_1^{PG})\lambda]$ and $D_2^{PG} \sim P[(p_1^{PG} - p_2^{PG})\lambda]$, respectively. Hence, the seller's expected revenue functions are:

$$\begin{cases} R_2^{PG} = p_2^{PG} \cdot E_{D_2^{PG}}\left[\min\left(D_2^{PG}, q\right)\right] - x_1\left(p_1^{PG} - p_2^{PG}\right) + \\ R^{PG} = p_1^{PG} \cdot E_{D_1^{PG}}[\min(D_1^{PG}, Q)] + E_{D_1^{PG}}(R_2^{PG*}) \end{cases} \tag{8}$$

Since we focus on the unknown market size and strategic consumer behavior, the numerical results are generated by the parameters $\{\alpha, \beta, \delta, \lambda, Q\}$. For the prior distribution of market size $\Lambda \sim \Gamma(\alpha, \beta)$ estimated by the seller before sales, the following definitions and meanings hold.

We define the mean $E(\Lambda) = \frac{\alpha}{\beta}$ as the initial estimate about the prior market size [29], where $\frac{\alpha}{\beta} = \lambda$ means the initial estimate is accurate; otherwise, it is not accurate. Additionally, the coefficient of variation $CV = \frac{\sqrt{Var(\Lambda)}}{E(\Lambda)} = \frac{1}{\sqrt{\alpha}}$ reflects the uncertainty of prior market size [61]. Moreover, a higher $CV$ indicates that the seller is more willing to update the prior distribution of the market size by demand learning [29,61]. Next, we explore how the initial estimate and uncertainty of the prior market size affect the performance of demand learning in two dynamic pricing alternatives.

The specific calculation process is as follows: By inserting the current state variables and parameters before the start of sales, such as $Q, \alpha, \beta, \lambda, \delta$, we solve the models in Section 4 and obtain the price decisions based on dynamic programming. Note that the first-stage realized sales volume is a random sample generated by the first-stage true demand distribution (i.e., the first-stage demand distribution under *PI* models). Finally, we generate 100 random first-stage sales samples and obtain the expected revenue with/without demand learning, as well as the optimal expected revenue $R^{PD*}$ (or $R^{PG*}$) in the *PI* model. Note that $R^{PD*}$ (or $R^{PG*}$) is the upper bound of the expected revenue for the original problem. We also define $b^{DP} = \frac{R^{DL} - R^{DN}}{R^{PD*}}$ and $b^{GP} = \frac{R^{GL} - R^{GN}}{R^{PG*}}$ to measure performance with demand learning versus that without demand learning under two alternatives of dynamic pricing, where a higher $b^{DP}$ (or $b^{GP}$) means that demand learning performs better.

### 5.1 Impact of Initial Estimate and Uncertainty on Demand Learning

The analysis here includes the following dimensions: First, we assume that true market size is fixed at $\lambda = 20$ and the seller's initial estimate of the prior market size $E(\Lambda) = \frac{\alpha}{\beta} \in \{10, 11, \cdots, 30\}$ to investigate how the initial estimate of the prior market size affects the results of dynamic pricing strategies. To eliminate the influence of $CV$ ($\frac{1}{\sqrt{\alpha}}$) on the models, we set $\alpha = \{1, 80\}$ so that $CV$ is consistent ($\alpha = 1$ means the high $CV = 1$ and $\alpha = 80$ means the low $CV = 0.1118$), and only change the value of $\beta$ to vary the initial estimate. Second, let the initial estimate $E(\Lambda) = \frac{\alpha}{\beta} = 20$ and the actual market size $\lambda$ is $10, 20,$ *and* $30$, respectively. We adjust the value of $\alpha$ and $\beta$ to make $CV$ vary from $0.1118$ to $1$ so that we can explore the impact of the uncertainty of prior market size on the results under dynamic pricing strategies. In addition, in both dimensions mentioned above, we take three levels initial inventory $Q \in \{10, 20, 30\}$ and the discount factor of consumer utility surplus $\delta = 0.8$. Some of these parameter values are based on Sen and Zhang [29] and Aviv et al. [57].

Figs 2 and 3 show the impact of the initial estimate and uncertainty on the performance of demand learning under the two dynamic pricing alternatives.

From these, we obtain the following Observation 1.

**Observation 1.** *(i) If no offering price guarantees in dynamic pricing, when the initial estimate is not underestimated ($E(\Lambda) \geq \lambda$), demand learning is slightly beneficial only if prior market size uncertainty is low; when the initial estimate is underestimated ($E(\Lambda) < \lambda$), demand learning always backfires, especially for high prior market size uncertainty.*

(ii) *For dynamic pricing with price guarantee, if the initial estimate is inaccurate ($E(\Lambda) \neq \lambda$), demand learning is superior and the benefit is more obvious under a more uncertain prior market size; meanwhile, when the initial estimate is relatively accurate ($E(\Lambda) = \lambda$), as the prior market size uncertainty increases, demand learning is inferior.*

(iii) *Demand learning has a more positive effect when the price guarantee is adopted for dynamic pricing.*

From Observation 1, if there is no a price guarantee, demand learning may be superior when prior market size uncertainty is relatively low. These findings differ from those of Sen and Zhang [29] and Talebian et al. [61], who do not consider strategic consumers. Our explanation is as follows. When the uncertainty of the prior market size is higher–the seller's willingness to update the market size is stronger, demand learning can create a higher first-stage purchasing threshold for strategic consumers, which further negatively affects the seller's revenue.

 

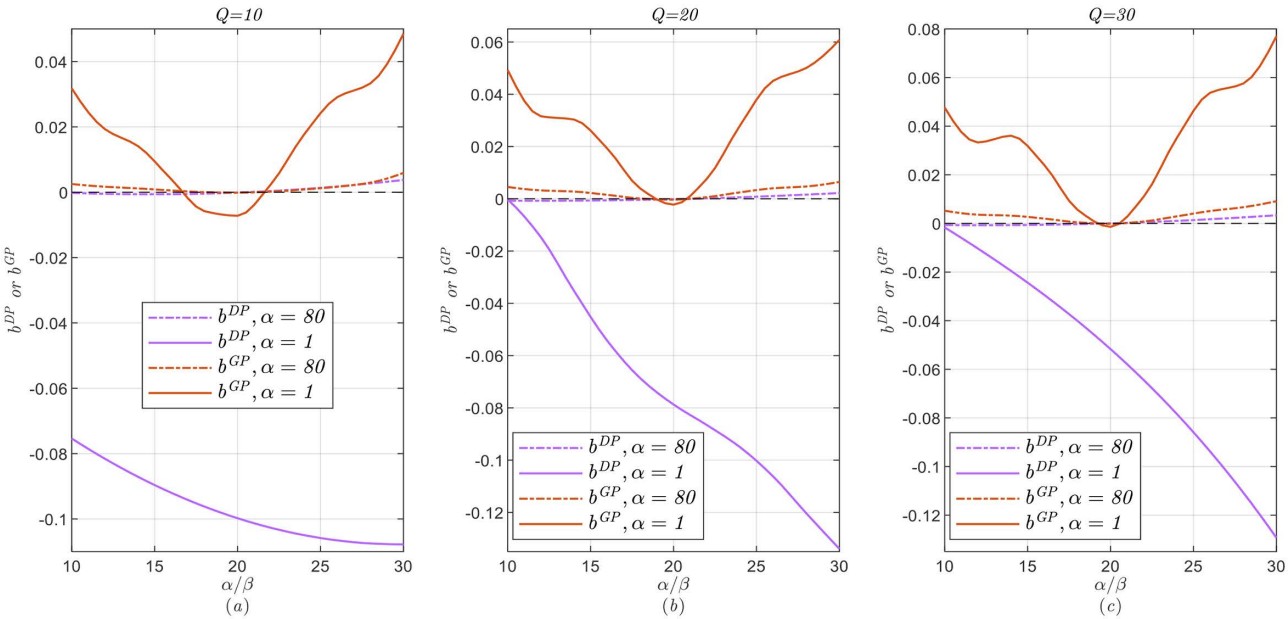

**Fig 2. The impact of the initial estimate on demand learning performance.**

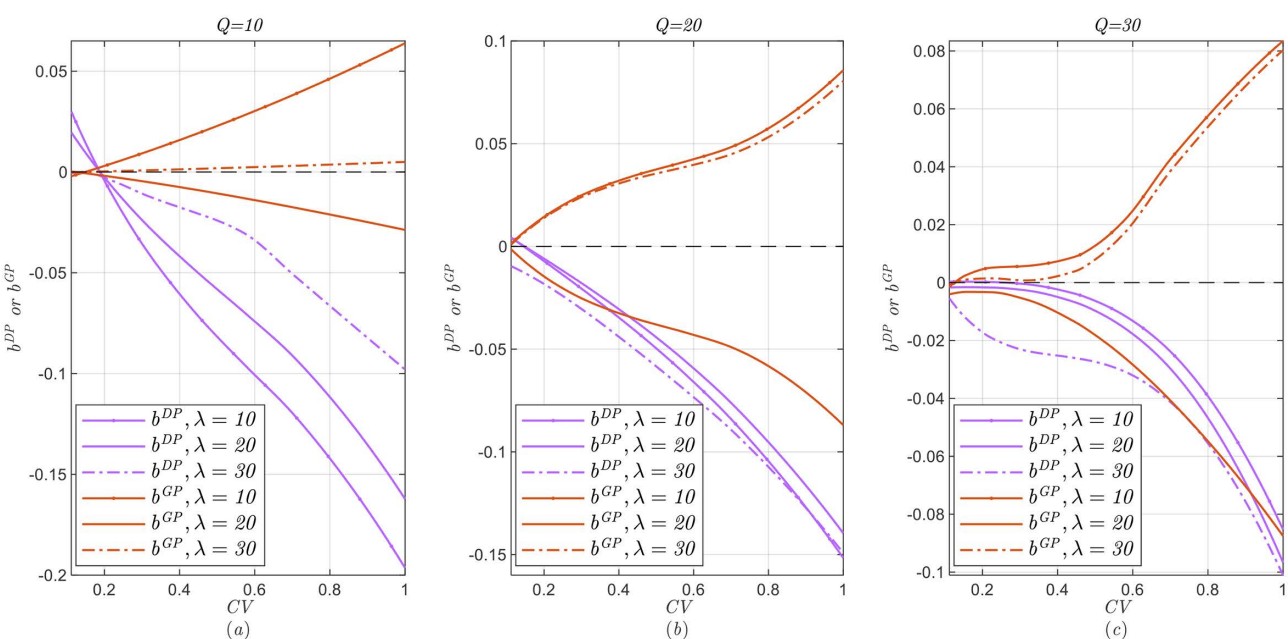

**Fig 3. The impact of prior market size uncertainty on demand learning performance.**

Interestingly, for dynamic pricing with price guarantee, the impacts of the initial estimate and prior market size uncertainty on demand learning performance are significantly different. Conversely, these results (Observation 1(ii)) are consistent with those of Sen and Zhang [29] and Talebian et al. [61]. This is because offering price guarantees can eliminate consumers' strategic behavior; therefore, demand learning can effectively correct estimates in the case of inaccurate initial estimate. Moreover, the higher the uncertainty of the prior market size, the more significant the correction effect of demand learning. However, for an accurate initial estimate, demand learning can lead to larger estimate bias under higher prior market size uncertainty.

Overall, due to irrelevant consumers' strategic behavior, demand learning remains beneficial in the case of offering price guarantees under certain conditions. Therefore, we believe that, when considering strategic consumers, for companies such as Groupon, Amazon, MediaMarkt, and Walmart, dynamic pricing with price guarantee and demand learning is recommended in most cases after new products are launched, especially for high uncertainty of prior market size.

## 5.2 Applicability of demand learning

According to the results in Section 5.1, we believe that demand learning is not always effective for the two dynamic pricing alternatives. Rather, it greatly depends on the prior distribution of the market size. We summarize the available conditions for demand learning under alternative dynamic pricing in Table 4. For dynamic pricing without price guarantee, demand learning is applicable only when the initial estimate is accurate or overestimated and the prior market size uncertainty is low. However, for dynamic pricing with price guarantee, demand learning is suitable when the initial estimate is inaccurate, particularly when the prior market size is highly uncertain.

## 6. Trade-off between dynamic pricing strategies adopting demand learning

Here, we investigate how Strategies *DL* and *GL* should be chosen consider the product sales period and strategic consumers, and illustrate managerial insights. Considering market conditions (the seller's initial estimate, uncertainty of prior market size, and true market size), consumers' strategic behavior, and initial inventory levels, a favorable dynamic pricing strategy is obtained for the seller by comparing the expected revenue.

In Section 5, we define the accuracy of the seller's estimation of the market size based on whether the initial estimate is equal to the true market size. In fact, determining the true market size in advance is difficult. Fortunately, we believe that the seller can judge the accuracy of the initial estimate from other perspectives, such as the product sales period. When a product is in the introduction period, the market size may be incorrectly estimated. In the mature period, the market size can be estimated relatively accurately. In addition, although dynamic pricing with price guarantee can effectively alleviate consumers' strategic waiting behavior, the seller may find that it is costly to pay the price-difference compensation to consumers who purchased at a high price in the first stage. Hence, whether offering a price guarantee always benefits the seller remains an unresolved issue. We need to analyze how to choose the two dynamic pricing strategies with demand learning (Strategies *DL* and *GL*) during different product sales periods considering the consumer's strategic level.

Table 4. Conditions of demand learning for two dynamic pricing alternatives.

| Uncertainty of prior market size | Dynamic Pricing without Price Guarantee | | Dynamic Pricing with Price Guarantee | |
|---|---|---|---|---|
| | Underestimating the market size ($E(\Lambda)<\lambda$) | Overestimating or accurate estimating the market size ($E(\Lambda)\geq\lambda$) | Underestimating or overestimating the market size ($E(\Lambda)\neq\lambda$) | Accurate estimating the market size ($E(\Lambda)=\lambda$) |
| Low | * | + | + | * |
| High | ** | * | ++ | ** |

Note: "+" means demand learning, "*" means no demand learning, and the number of symbols represents significance.

## 6.1 Introduction period of products considering consumer strategic level

When products first enter the market, because of a lack of historical sales data, the seller only acquires limited market information; thus, the prior distribution of the market size is usually incorrectly estimated. At this time, if the seller is optimistic (pessimistic) about the market, they may overestimate (underestimate) the market size.

Let the seller's initial estimate $\frac{\alpha}{\beta} = 20$, the actual market size $\lambda \in \{10, 30\}$, inventory $Q \in \{10, 20, 30\}$, $CV \in \{0.1118, 0.1581, 0.2236, 0.5, 0.7071, 1\}$, and consumer strategic level $\delta \in \{0.2, 0.4, 0.6, 0.8\}$. Tables 5 and 6 show the percentage improvement of revenue under Strategy $GL$ over Strategy $DL$ when market size is overestimated (optimistic) or underestimated (pessimistic), respectively. Note that $k = \frac{R^{GL}}{R^{DL}} - 1$ is defined and its value determines the choice of Strategies $DL$ and $GL$. Based on the results in Tables 5 and 6, we obtain Observation 2.

**Observation 2.** *When a product just enters the market, the trade-off between Strategies DL and GL is as follows.*

(i) *When the seller is optimistic about the market (see Table 5), Strategy GL is always preferred.*

(ii) *When the seller is pessimistic about the market (see Table 6), if consumers are strongly strategic (e.g., high $\delta$), Strategy GL is superior; otherwise, Strategy DL is preferred for weak strategic consumers (e.g., low $\delta$). In addition, with the rise of inventory level, strategic consumers with relatively low $\delta$ also need Strategy GL.*

Observation 2 provides a reference for choosing Strategies $DL$ and $GL$ for the seller who has just launched a product in the market. Regardless of whether the seller is optimistic about the market, that is, whether the market size is overestimated, Strategy $GL$ should be adopted in most cases for the introduction period of products, especially for strong strategic consumers.

## 6.2 Mature period of products considering consumer strategic level

When a product is in the mature period, owing to rich historical sales data and experience, the seller can easily obtain a relatively accurate initial estimate of the prior market size. We take the same parameters as Section 6.1 except for $\lambda = 20$. In this case, $k = \frac{R^{GL}}{R^{DL}} - 1$ is defined to determine whether to conduct Strategy $GL$. The results for $k = \frac{R^{GL}}{R^{DL}} - 1$ are illustrated in Table 7. Then, we obtain Observation 3.

**Table 5. Percentage improvement of Revenue under Strategy $GL$ over Strategy $DL$ when the seller is optimistic to the market during introduction period ($\frac{\alpha}{\beta} > \lambda = 10$).**

| Q | $\delta$ | CV | | | | | |
|---|---|---|---|---|---|---|---|
| | | 0.1118 | 0.1581 | 0.2236 | 0.5 | 0.7071 | 1 |
| 10 | 0.2 | 16.24% | 16.40% | 16.64% | 17.74% | 18.69% | 20.24% |
| | 0.4 | 19.80% | 20.20% | 20.80% | 23.61% | 26.08% | 30.27% |
| | 0.6 | 21.73% | 23.15% | 25.28% | 36.01% | 46.60% | 67.62% |
| | 0.8 | 29.66% | 31.70% | 34.77% | 50.89% | 67.80% | 104.79% |
| 20 | 0.2 | 17.56% | 17.47% | 17.36% | 17.19% | 17.40% | 18.25% |
| | 0.4 | 22.05% | 22.91% | 24.09% | 28.38% | 30.72% | 32.41% |
| | 0.6 | 32.43% | 33.14% | 34.25% | 40.59% | 47.67% | 63.24% |
| | 0.8 | 42.51% | 44.07% | 46.47% | 60.05% | 75.63% | 113.43% |
| 30 | 0.2 | 12.20% | 14.21% | 16.27% | 17.53% | 15.84% | 21.68% |
| | 0.4 | 21.33% | 21.19% | 22.10% | 27.75% | 25.72% | 35.88% |
| | 0.6 | 30.93% | 31.17% | 32.43% | 38.98% | 46.47% | 53.60% |
| | 0.8 | 43.08% | 43.41% | 44.93% | 54.44% | 65.25% | 91.57% |

Note: $k = \frac{R^{GL}}{R^{DL}} - 1$ is positive, indicating that Strategy $GL$ is superior. Otherwise, the Strategy $DL$ performs better.

**Table 6. Percentage improvement of Revenue under Strategy *GL* over Strategy *DL* when the seller is pessimistic to the market during introduction period ($\frac{\alpha}{\beta}<\lambda=30$).**

| Q | δ | CV | | | | | |
|---|---|---|---|---|---|---|---|
| | | 0.1118 | 0.1581 | 0.2236 | 0.5 | 0.7071 | 1 |
| 10 | 0.2 | −8.63% | −8.78% | −8.91% | −9.10% | −9.69% | −13.06% |
| | 0.4 | −4.82% | −4.82% | −4.65% | −5.31% | −6.27% | −7.79% |
| | 0.6 | −0.49% | −0.46% | −0.40% | −0.05% | 0.32% | 1.04% |
| | 0.8 | 4.42% | 4.46% | 4.53% | 5.22% | 6.19% | 8.38% |
| 20 | 0.2 | −7.75% | −7.02% | −6.29% | −6.11% | −7.27% | −6.60% |
| | 0.4 | −0.43% | −0.03% | 2.68% | 4.00% | 3.89% | 4.21% |
| | 0.6 | 10.07% | 10.65% | 14.23% | 15.98% | 16.64% | 25.36% |
| | 0.8 | 22.88% | 26.66% | 24.03% | 31.02% | 32.52% | 55.63% |
| 30 | 0.2 | −7.99% | −7.36% | −6.43% | −5.57% | −8.82% | −5.98% |
| | 0.4 | 2.12% | 2.08% | 2.64% | 3.01% | 0.83% | 4.20% |
| | 0.6 | 13.61% | 13.99% | 14.27% | 15.83% | 15.94% | 16.41% |
| | 0.8 | 27.66% | 28.04% | 28.27% | 29.96% | 30.27% | 37.73% |

Note: $k=\frac{R^{GL}}{R^{DL}}-1$ is positive, indicating that Strategy *GL* is superior. Otherwise, Strategy *DL* performs better.

**Table 7. Percentage improvement of Revenue under Strategy *GL* over Strategy *DL* in the maturity period of products ($\frac{\alpha}{\beta}=\lambda=20$).**

| Q | δ | CV | | | | | |
|---|---|---|---|---|---|---|---|
| | | 0.1118 | 0.1581 | 0.2236 | 0.5 | 0.7071 | 1 |
| 10 | 0.2 | −10.47% | −10.26% | −11.22% | −13.16% | −14.52% | −19.26% |
| | 0.4 | −7.46% | −7.23% | −8.19% | −9.92% | −11.31% | −13.30% |
| | 0.6 | −2.73% | −3.78% | −3.37% | −2.17% | 0.36% | 1.41% |
| | 0.8 | −0.57% | 0.95% | 1.99% | 4.53% | 6.60% | 11.56% |
| 20 | 0.2 | −0.57% | −0.58% | −1.35% | −5.62% | −2.83% | −13.17% |
| | 0.4 | 7.09% | 5.80% | 7.60% | 3.79% | 2.07% | −2.05% |
| | 0.6 | 18.47% | 17.64% | 16.60% | 13.83% | 13.61% | 16.67% |
| | 0.8 | 30.02% | 29.59% | 29.14% | 29.45% | 32.45% | 42.74% |
| 30 | 0.2 | 0.78% | 0.77% | 0.76% | 2.15% | −4.44% | −5.81% |
| | 0.4 | 10.31% | 10.29% | 10.26% | 12.24% | 5.06% | 4.69% |
| | 0.6 | 21.79% | 21.76% | 21.71% | 24.42% | 21.55% | 17.49% |
| | 0.8 | 35.77% | 35.72% | 35.63% | 39.30% | 36.89% | 41.57% |

Note: $k=\frac{R^{GL}}{R^{DL}}-1$ is positive, indicating that Strategy *GL* is superior. Otherwise, Strategy *DL* performs better.

***Observation 3.*** *In the maturity period of products ($\frac{\alpha}{\beta} = \lambda = 20$), Strategy GL should be considered when consumers are strongly strategic, especially for high uncertainty of prior market size. However, if consumers are weak strategic, Strategy DL can be adopted, especially for low inventory.*

Observation 3 demonstrates a dynamic pricing strategy during the product maturity period. The trade-off between Strategies *DL* and *GL* is similar to that in Observation 2(ii).

According to Observations 2 and 3, the Strategy *GL* is superior in most cases, especially for highly strategic consumers. In addition, according to the numerical results, the percentage improvement of revenue generated by strategy *GL* is remarkable. (Here $k \geq 2\%$ in most cases and the highest is $k = 113.4\%$. Feldman [78] states that in the air industry,

if revenue increases by 1%, profits will increase by 60%. And Robinson [79] reports a 1% increase in revenue, profits increase by 14.29%.) Strategy *DL* may be preferred only when consumers are weakly strategic and the inventory is low. This is reasonable. When consumers are less strategic, the discounted utility in the second stage is low. If the inventory is low, consumers will face a higher shortage risk in the second stage. Based on these two factors, consumers are more willing to purchase in the first stage. Hence, the seller can directly adopt dynamic pricing instead of offering price guarantees and considering compensating consumers in the second stage. Conversely, when consumers are more strategic, they are more likely to wait for a purchase in the second stage. In this case, the seller needs to offer a price guarantee to attract more consumers to buy immediately, where the increased revenue in the first stage is much higher than the price difference compensation in the second stage.

In practice, the magnitude of consumers' strategic level ($\delta$) is closely related to product types and attributes. The seller can obtain the value of $\delta$ and then consider whether to conduct price guarantee. For example, fresh or perishable products have a low $\delta$ and durable products have relatively high $\delta$, and fashion-like products are generally considered to be medium level of $\delta$ [57]. Thus, the seller finds it more reasonable to offer a price guarantee on durable and fashion-like products, which is consistent with business practices.

## 7. Extension

In the above model, the inventory level is exogenous and the seller's pricing decisions are primarily explored. Here, we expand the initial inventory level as a decision variable, and the seller must determine $Q$, $p_1$ and $p_2$ to maximize the expected profit. Here, only the unit cost of products ($c$) is considered; the other costs and salvage values are ignored.

In Strategy *DL*, the seller's expected profit is expressed as follows:

$$R_Q^{DL} = E_{D_1^{DL}} \left( p_1^{DL} \cdot \min \left( D_1^{DL}, Q \right) - cQ + \max_{p_2^{DL}} \left( p_2^{DL} \cdot E_{D_2^{DL}} \left[ \min \left( (Q - x_1)^+, D_2^{DL} \right) \right] \right) \right) \tag{9}$$

where $D_1^{DL} \sim NB(\alpha, \frac{\beta}{\beta + 1 - v_1^{DL}(p_1^{DL})})$, $D_2^{DL} \sim NB(\alpha + x_1, \frac{\beta + 1 - v_1^{DL}(p_1^{DL})}{\beta + 1 - p_2^{DL}})$, and $x_1$ is the demand realization in the first stage.

Similarly, we get the expected profit function of Strategy *GL*:

$$R_Q^{GL} = E_{D_1^{GL}} \left( p_1^{GL} \cdot \min \left( D_1^{GL}, Q \right) - cQ \right)$$
$$+ E_{D_1^{GL}} \left( \max_{p_2^{GL}} \left( p_2^{GL} \cdot E_{D_2^{GL}} \left[ \min \left( (Q - x_1)^+, D_2^{GL} \right) \right] - x_1 \left( p_1^{GL} - p_2^{GL} \right)^+ \right) \right) \tag{10}$$

where $D_1^{GL} \sim NB(\alpha, \frac{\beta}{\beta + 1 - p_1^{GL}})$, $D_2^{GL} \sim NB(\alpha + x_1, \frac{\beta + 1 - p_1^{GL}}{\beta + 1 - p_2^{GL}})$.

To compare the performance of demand learning, *perfect information (PI) model* (i.e., the true market size $\lambda$ is known to the seller) is still chosen as the benchmark, where the expected profits are denoted as $R_Q^{PD*}$ and $R_Q^{PG*}$, respectively.

Specifically, we use the following hierarchical search procedure [76] to solve the models. Here, we take the Eq. (9) as an example. First, consider a set of inventory $Q \in \{0, 1, 2, \cdots\}$. We then solve for the price and expected profit at a particular inventory value according to the calculation process in Section 4. Finally, we search for the inventory by solving the expression $R_Q^{DL*} = \max(R_Q^{DL})$; that is, $Q^{DL*} = argmax_Q R_Q^{DL}$. The solutions for the other models are obtained using similar steps.

Similarly, we set $b^{DP} = \frac{R_Q^{DL} - R_Q^{DN}}{R_Q^{PD*}}$ and $b^{GP} = \frac{R_Q^{GL} - R_Q^{GN}}{R_Q^{PG*}}$ to measure the performance of demand learning in Strategies *DL* and *GL*. Let $\frac{\alpha}{\beta} = 20$, $CV \in \{0.1118, 0.2236, 1\}$, $\delta \in \{0.2, 0.4, 0.6, 0.8\}$, $\lambda \in \{10, 20, 30\}$, and $c \in \{0.1, 0.2\}$. The results are reported in Tables B1, B2, and B3 in Appendix B in S1 Appendix. Then, we obtain Observation 4.

*Observation 4. (i) Under dynamic pricing without price guarantee, demand learning is beneficial only when the initial estimate is accurate or overvalued ($\frac{\alpha}{\beta} \geq \lambda$), and prior market size uncertainty is low. Meanwhile, under dynamic pricing with price guarantee, demand learning is popular when the initial estimate is inaccurate ($\frac{\alpha}{\beta} \neq \lambda$), especially for a highly uncertain prior market size.*

*(ii) Strategy DL is always detrimental. Further, Strategy GL is profitable in the following case: (a) the initial estimate is overvalued, and (b) the initial estimate is undervalued and consumers are more strategic.*

Observation 4 summarizes the value of demand learning, and the trade-off between Strategies *DL* and *GL* when the inventory level is a decision variable. Compared with the results when inventory is exogenous, for dynamic pricing alternatives, the applicable conditions of demand learning are consistent. However, one difference exists: there is no opportunity to adopt Strategy *DL*.

## 8. Conclusion

Based on Bayesian updating, this study develops a set of two-stage dynamic pricing models with and without price guarantee to explore the impact of demand learning on seller decisions towards strategic consumers. The main conclusions are as follows.

### 8.1 Managerial insights and theoretical implications

The findings of our study are important for guiding sellers to price products for strategic consumers through demand learning. The managerial insights are summarized as follows:

First, we demonstrate the applicability of demand learning under dynamic pricing with or without price guarantee for strategic consumers. Under dynamic pricing without price guarantee, the applicable conditions for demand learning are counterintuitive. Specifically, demand learning may yield a marginal benefit only when prior market size uncertainty is low; that is, the seller is less willing to update the prior market size. Otherwise, this strategy can backfire (see Observation 1(i)). This is because demand learning can result in a higher purchase threshold for strategic consumers in the first stage for more uncertain prior market size, which harms the seller's revenue. Nevertheless, for dynamic pricing with price guarantee, demand learning is valid due to significant correction effect when products just enter the market and with a highly uncertain prior market size. Therefore, for companies such as Groupon, Amazon, MediaMarkt, and Walmart, dynamic pricing with price guarantee and demand learning (i.e., Strategy *GL*) is preferred in most cases after new products are launched, especially for highly uncertain prior market size. However, demand learning is also detrimental for offering price guarantees under certain scenarios. For example, if products are in the mature stage, demand learning is invalid (see Observation 1(ii)).

Second, we illustrate the advantages and disadvantages of implementing price guarantees in dynamic pricing for strategic consumers. Compared to no price guarantee, if a price guarantee is offered, demand learning has more applicable scenarios and is more effective (see Observation 1(iii)). In addition, when consumers' strategic behavior is strong, offering a price guarantee can significantly benefit the seller's revenue. However, if consumers' strategic behavior is weak, it is no longer effective because of the price difference compensation losses (see Observations 2 and 3). Our results are consistent with the business practices. For instance, JD.com mainly offers price guarantees for TV sets, refrigerators, computers, and other home appliances, but not for fresh goods because consumers show a strong strategic wait for such durable products. In contrast, dynamic pricing with/without price guarantee is available because consumers have a medium level of strategic behavior for fashion-like products, including smartphones, branded apparel, and cosmetics.

Finally, we provide guidance for sellers on the trade-off between two dynamic pricing alternatives with demand learning. To develop a better pricing strategy, a seller should consider the product sales period, uncertainty of prior market size, level of consumers' strategic behavior, and inventory. If a product is in the introduction period and the seller is optimistic about the market, dynamic pricing with price guarantee and demand learning (i.e., Strategy *GL*) is always

 

preferable (Observation 2(i)). When the product in the introduction period and the seller is pessimistic about the market, or if the product is in the mature period, Strategy *GL* is profitable in most cases, especially for strong strategic consumers. Dynamic pricing without price guarantee and with demand learning (i.e., Strategy *DL*) should be chosen only when consumers' strategic behavior is weak or the initial inventory is low (see Observations 2(ii) and 3).

Next, this study also has significant theoretical implications. The literature emphasizes that strategic consumers negatively affect dynamic pricing without price guarantee and that offering price guarantees can effectively eliminate strategic consumer behavior. Meanwhile, demand learning is popular in practice and academic research because it can help a seller increase revenue by learning early data to dynamically generate a more accurate price response. However, few studies have considered these factors simultaneously, especially focusing on the applicable conditions of demand learning for dynamic pricing with and without price guarantee. This study enriches the literature in this aspect, and provides a basis for how demand learning counters the strategic behavior of consumers for dynamic pricing with and without price guarantee. Moreover, we find a trade-off between dynamic pricing strategies that adopt demand learning.

### 8.2 Limitations and future research

There are three shortcomings and future research directions in our paper. First, we assume that consumers are completely rational. However, consumers are also likely to be bounded rational, whose decision is satisfactory rather than necessarily optimal [80–82]. In the future, bounded rational consumers can be considered in our model to explore their impact on research results. Second, we do not consider competing sellers and the reference price effect of strategic consumers [83–84]. These issues can be further explored in the future. Therefore, dynamic pricing with demand learning can be applied to the above scenarios to further analyze the value of demand learning. Finally, there might be interesting findings if the models are extended to supply chain settings and applied to emerging online patterns (advanced selling, livestreaming sales, etc.).

## Supporting information

**S1 Appendix. Plos one-S1 Appendix.**
(DOCX)

**S1 Fig. Sequence of events.**
(TIF)

**S2 Fig. The impact of the initial estimate on demand learning performance.**
(TIF)

**S3 Fig. The impact of prior market size uncertainty on demand learning performance.**
(TIF)

## Acknowledgments

The authors would like to express grateful to editors and reviewers of this paper for their insightful comments and suggestions.

## Author contributions

**Conceptualization:** Meixian Tang, Junfeng Tian.

**Formal analysis:** Meixian Tang, Junfeng Tian, Jinxia Kong, Zhenzhen Ren, Jinsong Tian.

**Funding acquisition:** Junfeng Tian.

**Investigation:** Meixian Tang.

**Methodology:** Meixian Tang, Junfeng Tian.

**Supervision:** Junfeng Tian.

**Validation:** Jinxia Kong, Zhenzhen Ren, Jinsong Tian.

**Writing – original draft:** Meixian Tang.

**Writing – review & editing:** Meixian Tang, Junfeng Tian.

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
