## [Decision Letter · Decision Letter 0]

17 Nov 2025

Dear Dr. Tian,

Thank you for submitting your manuscript to PLOS ONE. After careful consideration, we feel that it has merit but does not fully meet PLOS ONE’s publication criteria as it currently stands. Therefore, we invite you to submit a revised version of the manuscript that addresses the points raised during the review process.

We look forward to receiving your revised manuscript.

Kind regards,

Babu George

Academic Editor

PLOS ONE

Journal Requirements:

**Additional Editor Comments:**

Please give attention to the comments made by reviewer II. I understand if all recommended changes are not feasible (might require additional fundamental research and restructuring): in such cases, please explain why certain comments were not addressed well.

Reviewers' comments:

Reviewer's Responses to Questions

**Comments to the Author**

1. Is the manuscript technically sound, and do the data support the conclusions?

Reviewer #1: Yes

Reviewer #2: Yes

2. Has the statistical analysis been performed appropriately and rigorously?

Reviewer #1: Yes

Reviewer #2: Yes

3. Have the authors made all data underlying the findings in their manuscript fully available?

Reviewer #1: Yes

Reviewer #2: No

4. Is the manuscript presented in an intelligible fashion and written in standard English?

Reviewer #1: Yes

Reviewer #2: Yes

Reviewer #1: I am attaching my review report in PDF format for your consideration. The document contains my detailed assessment of the manuscript, including comments on its theoretical contribution, methodology, empirical analysis, and overall clarity of presentation.

Reviewer #2: Please see the review attached.

Dynamic Pricing Strategies towards Strategic Consumers under Demand Learning

PONE-D-25-19671

I have carefully reviewed this manuscript; the following are my comments.

Comments and Suggestions

1. The manuscript makes a valuable contribution by addressing a clearly identified and underexplored research gap, as effectively summarized in Table 1. The simultaneous integration of demand learning, strategic consumer behavior, and price guarantees is a significant and timely endeavor that adds considerable depth to the existing literature.

2. A key model assumption—that sellers and consumers share a common prior and homogeneously update their beliefs via Bayesian learning (Section 3.3)—deserves further consideration. While this is a common and useful theoretical simplification, it may not fully capture real-world consumer behavior. Consumers often operate with limited information and may rely on heuristic decision-making rather than sophisticated Bayesian updates. I recommend that the authors explicitly acknowledge this as a limitation in the dedicated section and briefly discuss how incorporating heterogeneous priors or bounded rationality could influence their findings. This would not only strengthen the paper's self-awareness but also provide a compelling direction for future research.

3. While the numerical results are presented clearly as percentage changes, the economic significance of these differences could be strengthened. For instance, it is unclear whether a 1-2% revenue improvement would justify the operational shift to a new pricing strategy in practice. To enhance the impact of these findings, the authors could briefly contextualize the results by discussing the absolute revenue implications or commenting on the practical significance of the reported percentage changes from a managerial perspective.

4. There appears to be an inconsistency in the definition of the performance metric k in Section 6. The text defines it as a function of revenue (k = R_GL / R_DL - 1), which is the appropriate measure for strategy comparison. However, the headers and notes for Tables 4, 5, and 6 refer to it as a function of price (p_GL / p_DL - 1). This seems to be a typographical error. I recommend standardizing this to the revenue-based definition throughout the manuscript to avoid confusion.

5. The manuscript is generally well-written. To further enhance its clarity and academic tone, a careful proofread to polish minor grammatical points is recommended. For example:

(i) "demand learning brings slight benefits" could be refined to "demand learning yields only marginal benefits".

(ii) "is counterproductive for offering a price guarantee" could be rephrased to "can be counterproductive when a price guarantee is in place" or "undermines the effectiveness of a price guarantee".

6. As a summary suggestion, I strongly advise the authors to incorporate a discussion on the limitation of the homogeneous, rational consumer belief assumption into Section 8.2 (Limitations and future research). Elaborating on this point will significantly improve the manuscript's rigor and scope for future work.

Once my comments are addressed, the manuscript can be reconsidered. Good luck!

**Do you want your identity to be public for this peer review?** For information about this choice, including consent withdrawal, please see our Privacy Policy

Reviewer #1: No

Reviewer #2: **Yes: ** Fahad Ali

---

## [Author Response · Author response to Decision Letter 1]

15 Dec 2025

Authors’ response to reviewers’ for “Dynamic Pricing Strategies towards Strategic Consumers under Demand Learning”

Dear Editors and Reviewers:

We are deeply grateful for the opportunity to revise our manuscript entitled “Dynamic Pricing Strategies towards Strategic Consumers under Demand Learning” (ID: PONE-D-25-19671) by Meixian Tang, Junfeng Tian et al.

We sincerely thank the editors and all reviewers for their insightful comments, which have been invaluable in enhancing the quality of our manuscript. We have studied these suggestions carefully and made point-by-point corrections and improvements which are marked in red in the revised manuscript. We hope that these revisions meet with approval.

Yours sincerely,

The authors

A point-by-point response to the reviewers’ comments

Review comments from reviewer 1

Major concerns:

1.The topic is interesting, but the paper lacks a solid theoretical framework. The results need more explanation, and what is the economic meaning behind them. It is suggested to add any “theory” about competition intensity (multi-seller market power asymmetry) and reference price / fairness perception of early buyers which explains the rational of relationship between the competition intensity (multi-seller market power asymmetry) and the reference price / fairness perception of early buyers, what kind of connections should one expect? What direction should any causality have?

Our response: We greatly appreciate your back. We apologize that we cannot obtain more theoretical results due to the complexity of the model structure. Therefore, in section 4.5, we focus on explaining the reasons for the lack of analytical solutions. In addition, thank you very much for your suggestion to consider seller competition and reference price effect. However, due to the complexity of the model and the limited ability of the authors, we are currently unable to directly obtain the relationship between them. Therefore, in the revised version, we have placed this suggestion in section 8.2 as a limitation and future research direction of this paper, which is seen in Section 8.2 on page XX. We hope our modifications receive your understanding and recognition.

2.The literature review should be presented though in a clearer manner how the present study differentiates from existing research and makes a step towards better understanding the research problem. I recommend the authors to consult the following survey and empirical papers to contextualize your findings. This should help the readers to understand the novelty of your work (use the following references which are updated(recent) in the literature review section):

Dimitriadis, K. A., Koursaros, D., & Savva, C. S. (2025). Exploring the Dynamic Nexus of Traditional and Digital Assets in Inflationary Times: The Role of Safe Havens, Tech Stocks, and Cryptocurrencies. Economic Modelling, 107195.

Dimitriadis, K. A., Koursaros, D., & Savva, C. S. (2024). The influential impacts of international dynamic spillovers in forming investor preferences: a quantile-VAR and GDCC-GARCH perspective. Applied Economics, 1-21.

Our response: We thank you for this comment. In the revised manuscript, we have referred to the writing style of the two references you recommended, added a section, and polished the differences and contributions of our paper to the literature. The specific modifications refer to the red text in in Section 2.3 on page xx. We hope to meet your approval.

3.Also, the paper needs to improve the quality of the results section and how to present things (to be more explicit in discussing results, not so extensive, use the following article in the references section based on this: https://doi.org/10.1016/j.irfa.2024.103693).

Our response: We appreciate your suggestion. In the revised manuscript, we referred to the writing form of the literature you recommended, and made corresponding modifications in the conclusion section. Please refer to the red text in paragraph 1 of Section 8, paragraphs 2-4 of Section 8.1, and Section 8.2 for specific modifications.

4.The authors should give more emphasis on the importance of their and should also be clearer in the conclusion section. Overall, this paper is interesting and well-structured, and it is my belief that it will be appealing for the readers of the journal. Furthermore, the abstract it would be better to be one whole paragraph which includes the most important result, the innovation of the research article and what does this article study.

Our response: We acknowledge your feedback. In the revised version, we carefully read the literature you recommended and polished the importance of research questions and conclusions of the article. In addition, our abstract is presented in the form of one whole paragraph, which includes the research content, research methods, important conclusions. Please refer to the red text in the Abstract on page 2, the 4th paragraph of Section 1 on page XX. We hope our explanations and revisions can meet with your approval.

5.The paper should clearly address the following: What is the research question? Why is this question important and interesting? How did they approach the problem? and why is their empirical strategy persuasive?

Our response: Thank you for your suggestions. Our response is as follows:

As for research questions: in the Section introduction, we first state the benefits and importance of implementing dynamic pricing with demand learning through literature and businesses examples (Fisher, 2009; Cheung et al., 2017; Naceva, 2024; Mailmodo, 2024; Dowling, 2023). Second, we emphasize the presence of strategic consumers and the necessity of implementing price guarantees based on dynamic pricing, as well as corresponding examples (e.g., JD.com, Dell (Williams, 2020), The Home Depot (Ritterman, 2021) and The Good Guys (Coates, 2022)). It is evident that dynamic pricing with price guarantee has gradually become a popular pricing strategy. However, it is not sure whether dynamic pricing with price guarantee always brings an increase in revenue, and how demand learning performs in dynamic pricing with price guarantee. Therefore, we have summarized four research questions for this article, which is shown in the 5th paragraph of Section 1 on page XX.

Regarding the importance of research questions: on the one hand, there is currently no research analyzing whether demand learning has superiority in dynamic pricing with price guarantees when facing strategic consumers. On the other hand, we believe that the questions have practical significance and can provide guidance for sellers' strategic choices.

For the research methodology used in this article: We illustrated the decision sequence in the text to state our research methods more intuitively, as detailed in the red text of third paragraph om Section 3.2 on page xx.

The reason why our empirical strategies are persuasive: First, our numerical results obtained through theoretical analysis are reasonable and effective. Second, the strategies based on numerical results are consistent with real-life business cases. Please refer to Tables 4, 5, and 6 in the manuscript, as well as the last two paragraphs of Section 6.2.

We hope our explanations and modifications can meet your approval.

The references are as follows.

Fisher M. OR FORUM-Rocket Science Retailing: The 2006 Philip McCord Morse Lecture. Operations Research. 2009; 57(3): 527-540.

Cheung WC, Simchi-Levi D, Wang H. Technical Note—Dynamic Pricing and Demand Learning with Limited Price Experimentation. Operations Research. 2017; 65(6): 1722-1731.

Naceva N. The Ultimate Guide to Amazon Dynamic Pricing Strategy in 2024 [Internet]. 2024 May 15 [Cited 2024 Sep 22]. Available from: https://influencermarketinghub.com/amazon-dynamic-pricing/.

Mailmodo. 6 Dynamic Pricing Examples From Different Industries [Internet]. 2024 May 22 [Cited 2024 Sep 22]. Available from: https://www.mailmodo.com/guides/dynamic-pricing-examples/.

Dowling L. The Future of E-commerce: Dynamic Pricing Strategies Powered by AI [Internet]. 2023 Oct 2 [Cited 2024 Sep 22]. Available from: https://pathmonk.com/e-ommerce-dynamic-pricing-strategies-powered-by-ai/.

Williams G. 10 Retailers That Offer Price Matching [Internet]. 2020 Nov 18 [Cited 2024 Sep 22]. Available from: https://money.usnews.com/money/personal-finance/spending/articles/what-is-price-matching.

Ritterman L. Home Depot Price Match & Price Adjustment Policy [Internet]. 2021 Dec 27 [Cited 2024 Sep 22]. Available from: https://pricematchguarantee.net/home-depot-price-match/.

Coates M. Get your Good Guys Price Match now [Internet]! 2022 Nov 16 [Cited 2024 Sep 22]. Available from: https://thechampagnemile.com.au/good-guys-price-match-guide/.

6.Does the paper clearly justify when demand learning improves revenue and when it actually hurts revenue, especially under a price guarantee?

Our response: Thank you for your comment. In the manuscript, we justified when demand learning improves revenue and when it hurts revenue under two dynamic pricing alternatives, which is shown in Observation 1 of Section 5.1 and Table 3 of Section 5.2 on page xx. We hope our response meets with your approval.

7.Are the assumptions about consumer behaviour (uniform valuations, common knowledge, rational expectations) realistic enough for the conclusions to be valid in real markets?

Our response: Thank you for your constructive suggestion. Our assumption about consumer behavior is indeed not completely consistent with the real world. Therefore, in the revised manuscript, we will relax this assumption and propose the combination of bounded rational consumers and our models as a future research direction. The corresponding modifications are the red text in section 8.2 on page xx, and we hope these modifications meet with your approval.

8.How robust are the numerical results to changes in key parameters like inventory level, demand uncertainty, and consumer patience (δ)?

Our response: Thank you very much for your comment. We have carefully examined the numerical results in the manuscript, and the specific conclusions are as follows:

1. Our parameter values are reasonable and comprehensive, among which the actual market size λ is 10,20, and 30, the seller's initial estimate of the prior market size E(Λ)=αβ∈{10,11,⋯,30}, the coefficient of variation CV varies from 0.1118 to 1, initial inventory Q∈{10,20,30} and consumer strategic level δ∈{0.2,0.4,0.6,0.8}. Some of these parameter values are based on Sen and Zhang (2009) and Aviv et al. (2019).

2. In specific scenarios, when we conduct sensitivity analysis of key parameters, even if other parameter values are changed, the trend of numerical results is mostly consistent.

3. Our calculation process and indicators are standardized and referenced. Specifically, the calculation process can refer to Lin (2005) and Talebian et al. (2014), and the indicators bDP=RDL−RDNRPD∗ and bGP=RGL−RGNRPG∗ described in Section 5 follow to Sen and Zhang (2009) and Zhang et al. (2020). The indicator k=RGLRDL−1 in Section 6 refers to Shum et al. (2016).

Therefore, we believe that the numerical results are robust. I hope our response will receive your approval.

The references are as follows.

Sen A, Zhang AX. Style goods pricing with demand learning. European Journal of Operational Research. 2009; 1196: 1058-1075.

Aviv Y, Wei MM, Zhang, F. Responsive Pricing of Fashion Products: The Effects of Demand Learning and Strategic Consumer Behavior. Management Science. 2019; 65(7): 2982-3000.

Lin KY. Dynamic pricing with real-time demand learning. European Journal of Operational Research. 2006; 174: 522-538.

Talebian M, Boland N, Savelsbergh MW. Pricing to Accelerate Demand Learning in Dynamic Assortment Planning for Perishable Products. European Journal of Operational Research. 2014; 237: 555-565.

Shum S, Tong S, Xiao T. On the Impact of Uncertain Cost Reduction When Selling to Strategic Customers. Management Science. 2016; 63(3): 843-860.

Zhang M, Ahn HS, Uichanco J. Data-Driven Pricing for a New Product. Operations Research. 2022; 70(2): 847-866.

9.Do the authors convincingly explain how a seller should choose between “with price guarantee” and “without price guarantee” strategies at different product life-cycle stages?

Our response: Thanks for your comment. In the manuscript, we analyzed the reasons behind the seller making trade-offs between Strategy DL (dynamic pricing without price guarantee and with demand learning) and GL (dynamic pricing with price guarantee and demand learning), as detailed in the last two paragraphs of Section 6.2 (the red text). I hope our explanation will receive your understanding and approval.

Review comments from reviewer 2

Major concerns:

1.The manuscript makes a valuable contribution by addressing a clearly identified and underexplored research gap, as effectively summarized in Table 1. The simultaneous integration of demand learning, strategic consumer behavior, and price guarantees is a significant and timely endeavor that adds considerable depth to the existing literature.

Our response: We greatly appreciate your recognition of the format in Table 1. Your approval has been incredibly encouraging and motivating.

2.While the numerical results are presented clearly as percentage changes, the economic significance of these differences could be strengthened. For instance, it is unclear whether a 1-2% revenue improvement would justify the operational shift to a new pricing strategy in practice. To enhance the impact of these findings, the authors could briefly contextualize the results by discussing the absolute revenue implications or commenting on the practical significance of the reported percentage changes from a managerial perspective.

Our response: Thank you for your comments. Our results are presented in percentage, which is consistent with Sen and Zhang (2009) and Aviv et al. (2019), Lin (2005) and Talebian et al. (2014), Shum et al. (2016). Besides, we believe that a 1% increase in revenue can significantly increase profits. Specifically, Feldman (1990) reports that in the airline industry, an increase of 1% in revenue is equivalent to 60% increase in profit; and Robinson (2021) reports a 1% increase in revenue, profits increase by 14.29%. Moreover, In the numerical results of Section 6, only a small amount of |k| is less than 2%, most of |k| exceed 2%, and the highest is 113%, indicating that changing the pricing strategy can increase the expected revenue by more than double the original. Therefore, we believe that it is necessary and worth to justify pricing strategies in most cases. Finally, based on your suggestion, we have briefly explained the practical significance of revenue percentage changes from a managerial perspective. Please refer to the red text in the third paragraph of section 6.2 on page xx for specific modifications.

The references are as follows.

J.M. Feldman, Fares: To raise or not to raise. Air Transport World 27 (6) (1990) 58–59.

3.There appears to be an inconsistency in the definition of the performance metric k in Section 6. The text defines it as a function of revenue (k = R_GL / R_DL - 1), which is the appropriate measure for strategy comparison. However, the headers and notes for Tables 4, 5, and 6 refer to it as a function of price (p_GL / p_DL - 1). This seems to be a typographical error. I recommend standardizing this to the revenue-based definition throughout the manuscript to avoid confusion.

Our response: We appreciate your feedback. In the revised version, we have replaced the indicator k=RGLRDL−1 in the notes of Tables 4, 5, and 6 with k=RGLRDL−1, and we bolded them for readers to better browse the manuscript. Corresponding revisions are seen the red text in Tables 4, 5, and 6 of Sections 6.1-6.2 on pages xx .

4.The manuscript is generally well-written. To further enhance its clarity and academic tone, a careful proofread to polish minor grammatical points is recommended. For example:

(i) "demand learning brings slight benefits" could be refined to "demand learning yields only marginal benefits".

(ii) "is counterproductive for offering a price guarantee" could be rephrased to "can be counterproductive when a price guarantee is in place" or "undermines the effectiveness of a price guar

---

## [Editor Report · Decision Letter 1]

16 Dec 2025

Dynamic Pricing Strategies towards Strategic Consumers under Demand Learning

PONE-D-25-19671R1

Dear Dr. Tian,

We’re pleased to inform you that your manuscript has been judged scientifically suitable for publication and will be formally accepted for publication once it meets all outstanding technical requirements.

Kind regards,

Babu George

Academic Editor

PLOS One
---

## [Editor Report · Acceptance letter]

PONE-D-25-19671R1

PLOS One

Dear Dr. Tian,

I'm pleased to inform you that your manuscript has been deemed suitable for publication in PLOS One. Congratulations! Your manuscript is now being handed over to our production team.

Kind regards,

on behalf of

Dr. Babu George

Academic Editor

PLOS One